# The Mediating Role of Sleep Quality in the Relationship between Negative Emotional States and Health-Related Quality of Life among Italian Medical Students

**DOI:** 10.3390/ijerph20010026

**Published:** 2022-12-20

**Authors:** Matteo Carpi, Annarita Vestri

**Affiliations:** 1Department of Psychology, Sapienza University of Rome, 00185 Rome, Italy; 2Department of Public Health and Infectious Diseases, Sapienza University of Rome, 00185 Rome, Italy

**Keywords:** sleep quality, psychological distress, health-related quality of life, medical students, mediation analysis

## Abstract

Sleep problems have been shown to be related to adverse outcomes concerning physical and mental well-being. Furthermore, mental health issues and sleep problems were reported to be highly prevalent among medical students and physicians, and were found to be associated with worse academic and clinical performance in these populations. This study aims to investigate the prevalence of poor sleep to examine the associations between sleep quality and health-related quality of life (HRQoL), and to explore the possible mediating role of sleep in the relationship between psychological distress and HRQoL itself in a sample of medical and dental students attending a large Italian university. Participants (*n* = 407, mean age: 24.2 ± 2.4) answered an online questionnaire comprising the 21-item Depression Anxiety Stress Scale, the Pittsburgh Sleep Quality Index, and the Short Form-12 health survey. Up to 62% of the participants reported poor sleep quality. Controlling for psychological distress, sleep quality components were found to be associated with physical and mental HRQoL. Mediation analysis showed that overall sleep quality mediated all the single associations between anxiety, depression, and stress and HRQoL. These preliminary findings suggest that the quality of sleep is important for the well-being of medical students and that targeting sleep issues in this academic population may be beneficial.

## 1. Introduction

Medical education is taxing and emotionally demanding, and medical students are exposed to a high academic and clinical workload that may result in elevated stress impacting on their general well-being [1]. Indeed, mental health problems and burnout are consistently reported among physicians in training [2,3,4] and apparently perpetuate after employment, with an elevated prevalence of distress and depression [5,6], and a higher suicide rate in comparison with the general population [7]. Several systematic reviews highlighted significant levels of depression, anxiety, and psychological distress in medical students across different cultural contexts [8,9,10,11], and recent large meta-analyses yielded prevalence estimates of 27.2% for depression [12] and 33.8% for anxiety [13]. Additionally, these estimates might be even increased after the outbreak of COVID-19 [14] which arguably exerted a considerable impact on both university students’ and healthcare workers’ mental health [15,16].

Sleep quality among medical students has also been investigated, with findings showing a higher prevalence of sleep disturbances such as insomnia and excessive sleepiness in comparison with both the general population and non-medical university students [17]. As addressed in previous studies [18,19], more than 50% of aspiring physicians report poor sleep quality according to the Pittsburgh Sleep Quality Index [20], and up to 58% report sleeping less than seven hours per day [18], which is the minimum recommended amount of sleep for adults and young adults [21]. Interestingly, according to the literature, sleep loss and poor sleep are also associated with worse academic performance in medical students [17,22,23]. 

The bidirectional relationship between sleep problems and psychological distress is well-documented and acknowledged [24,25], and both mental health and sleep quality have been shown to be associated with general health and quality of life in the general population as well as among university students [26,27,28,29,30]. Thus, the co-occurrence of sleep disturbances and psychological complaints is indeed an expected outcome, and it is not surprising that significant associations between sleep quality and anxiety, depression, and stress were observed among medical students [31,32,33,34]. However, in order to disentangle these relationships and better understand their implications, research investigating the possible directional processes and mechanisms in the associations between sleep, mental health, and well-being is needed.

In fact, a few studies employed mediation analysis [35,36] for this purpose. Mostly, they showed that sleep quality mediates the relationships between several psychological and health-related variables in different populations, such as depression and quality of life in Portuguese older adults [37], psychological distress and quality of life in veterans with type 2 diabetes [38], stress and health-related quality of life among pregnant women [39], and perceived stress and depression among older Chinese people [40]. However, alternative paths of relationships have also been examined (e.g., anxiety was shown to mediate the relationship between perceived stress and sleep quality in the Chinese general population [41], whereas perceived stress mediated the associations between sleep quality and both symptoms of anxiety and depression in a study conducted on nursing students [42]). In any case, evidence regarding university students and medical students in particular is scant. To our knowledge, only one study investigated the mediating role of insomnia symptoms in the relationship between perceived stress and depression in a large sample of Chinese medical students after the onset of the COVID-19 pandemic [43]. The authors found that insomnia significantly mediated the relationship between stress and symptoms of depression, and consequently suggested that strategies and interventions to manage sleep problems should be developed and employed in order to prevent depression among medical students.

With the main objective of expanding the knowledge about the mechanisms implicated in the associations between sleep quality, psychological distress, and quality of life, this study aims at (1) exploring sleep quality and sleep habits among Italian medical students, (2) at investigating the associations between negative emotional states (i.e., anxiety, depression, and stress), specific sleep quality domains, and physical and mental health-related quality of life, and (3) at examining the mediating role of sleep quality in the relationships between negative emotional states and health-related quality of life.

## 2. Materials and Methods

### 2.1. Procedures and Participants

An invitation to participate with a description of the study’s objectives was delivered to students enrolled at Sapienza University of Rome through university social networks and mailing lists. Participation was voluntary and data were collected through an anonymous online cross-sectional survey delivered on the Google Forms platform from March 2021 to June 2021. Similar online procedures for collecting data have been widely used in psychological and sleep research [44,45,46] and facilitate obtaining responses from large samples in an affordable and reliable way. 

Four hundred and ten students attending a university course in Medicine and Surgery or Dentistry at Sapienza University of Rome responded to the survey and were included in this study. Participants who provided invalid responses were excluded (*n* = 3), and no further exclusion criteria were applied. Thus, the final sample comprised 407 students (mean age: 24.2 ± 2.4, 339 females, 66 males, and 2 who chose not to declare their sex). All participants provided online informed consent and the research procedures were approved by the competent Ethics Committee at Sapienza University of Rome (protocol number 0308/2021).

### 2.2. Measures

Information about participants’ academic position (course attended and year of study) and demographic data were obtained through dedicated questions, whereas negative emotional states, sleep quality, and health-related quality of life were assessed with the standardized measures described below. The measures’ reliability was evaluated through Cronbach’s alpha (*α*) [47] and McDonald’s omega (*ω*) [48] coefficients with reference to conventional criteria [49].

#### 2.2.1. Depression Anxiety Stress Scale-21

The 21-item Depression Anxiety Stress Scale (DASS-21) [50] was used to measure negative emotional states in three dimensions (namely, anxiety, depression, and stress). This instrument is composed of 21 items describing experiences, feelings, and thoughts related to negative affectivity and rated on a frequency scale from 0 (“never”) to 3 (“always”) referred to the previous week. Each dimension is measured by seven items, and its score is obtained by summing the single items’ responses.

The Italian version of the DASS-21 showed convergent validity with other measures of distress and psychological symptoms and good psychometric properties (*α* of 0.74, 0.82, and 0.85 for the anxiety, depression, and stress subscales, respectively, and *α* = 0.90 for the total score). In this study, satisfactory reliability was found for both the three subscales and the total score (*α* = 0.83 and *ω* = 0.84 for anxiety, *α* = 0.89 and *ω* = 0.90 for depression, *α* = 0.87 and *ω* = 0.88 for stress, and *α* = 0.93 and *ω* = 0.93 for the total score).

#### 2.2.2. Pittsburgh Sleep Quality Index

Sleep quality was measured with the Pittsburgh Sleep Quality Index (PSQI) [20]. A widely-used self-report questionnaire, the PSQI comprises 19 items with different response formats (5-point Likert scales and open-ended questions) concerning perceived sleep quality, habitual sleep schedules, and common sleep problems. The instrument measures seven dimensions (subjective sleep quality, sleep latency, sleep duration, habitual sleep efficiency, sleep disturbances, use of sleep medications, and daytime dysfunction) with aggregated scores between 0 and 3 obtained from specific items for each dimension. The sum of the dimensions’ scores yields a total score, with higher values corresponding to worse sleep quality. A cut-off of 5 demonstrated good sensitivity and specificity in discriminating between poor sleepers and good sleepers [20], and the Italian version of the PSQI showed good reliability (*α* = 0.83) and was able to distinguish patients with sleep disorders and healthy control.

In this study, the scale’s reliability was acceptable with *α* = 0.72 and *ω* = 0.74. Information about sleep variables (namely, total sleep duration, sleep latency, and sleep efficiency) was derived from the PSQI items.

#### 2.2.3. Short Form-12

Health-related quality of life (HRQoL) was evaluated through the Short Form-12 (SF-12) [51]. This 12-item questionnaire with different response scales (yes/no questions and three to six-point scales) measures eight dimensions of physical and mental functioning (physical activity, role and physical health, role and emotional state, mental health, physical pain, general health, vitality, and social activities). The weighted sums of the dimensions’ scores yield two indexes, the Physical Component Summary (PCS) and the Mental Component Summary (MCS), that account for the physical and mental aspects of health-related quality of life, respectively. Standardized scores (with a mean of 50 and a standard deviation of 10) are computed for the two components following the procedure described by Ware and collaborators [52] and allow for comparison with normative data.

In this study, the Italian adaptation of the SF-12 [53] was used, and the full 12-item instrument showed satisfactory reliability (*α* = 0.79, *ω* = 0.80).

### 2.3. Statistical Analyses

Statistical analyses were performed with IBM SPSS software (version 25.0, IBM Corp., Armonk, NY, USA). Descriptive statistics were obtained to explore sample characteristics. Categorical variables were summarized by counts and percentages with 95% confidence intervals (CIs), and means and standard deviations were computed for continuous variables. 

Mean differences in negative affect (i.e., DASS-21 anxiety, depression, and stress), sleep quality (PSQI total score), and health-related quality of life (SF-12 PCS and MCS scores) across sexes and years of study were examined through one-way analyses of variances (ANOVAs). In order to account for multiple comparisons, adjusted p-values were obtained for all the F statistics using the false discovery rate procedure proposed by Benjamini and Hochberg [54], and post-hoc comparisons were conducted using Hochberg’s GT2 test to control for different sample sizes across conditions [55]. The bivariate relationships between the investigated variables were evaluated with Pearson’s correlation coefficients.

Hierarchical multiple regression models were conducted to explore the unique associations between sleep quality components (PSQI subscales’ scores) and physical and mental health-related quality of life controlling for sex, age, and anxiety, depression, and stress symptoms. Furthermore, the mediating role of sleep quality in the relationship between negative emotional states and health-related quality of life was explored with two mediation models considering the DASS-21 anxiety, depression, and stress scores as antecedents, the SF-12 PCS and MCS as consequents, and the PSQI total score as mediator (see Figure 1). Mediation analyses were conducted using the PROCESS macro for SPSS (model 4) following the regression-based approach described by Hayes [36]. Accordingly, the indirect effect of a given variable X on a consequent variable Y through a mediator M was computed by multiplying the regression coefficient of X on M and the regression coefficient of M on Y.

Estimates for total, direct, and indirect effects were obtained, and a nonparametric bootstrapping procedure was adopted to evaluate the significance of the indirect effects [56]. Percentile bootstrap 95% confidence intervals each based on 5000 bootstrap samples were computed for the indirect effects (with seed = 22922), and an indirect effect was considered significant if its confidence interval did not comprise zero.

For all the other analyses performed, *p*-values below 0.05 were considered statistically significant.

## 3. Results

### 3.1. Participants’ Characteristics

Participants’ demographic, academic, and sleep characteristics are reported in Table 1, while sleep quality, anxiety, depression, stress, and health-related quality of life mean scores are shown in Table 2. The majority of the students involved in the study were female and the large part of them were enrolled in the final years of study (fifth year, sixth year, and supplementary years). Concerning sleep quality and duration, 61.9% (95% CI: 57.0–66.6%; *n* = 252) obtained a PSQI total score above 5 indicating overall poor sleep quality, and 33.7 (95% CI: 29.1–38.5%; *n* = 137) reported sleeping less than seven hours per night. The ANOVAs revealed no differences in negative emotional states, sleep quality, physical HRQoL, and mental HRQoL between females and males (all adjusted *p*-values ≥ 0.10), nor in sleep quality (adjusted *p* = 0.08) and physical HRQoL (adjusted *p* = 0.08) across years of study. On the other hand, significant differences across years of study were found in anxiety (*F*(6,400) = 2.66, *p* = 0.02, adjusted *p* = 0.03), depression (*F*(6,400) = 2.60, *p* = 0.02, adjusted *p* = 0.03), stress (*F*(6,400) = 2.77, *p* = 0.01, adjusted *p* = 0.02), and mental HRQoL (*F*(6,400) = 2.87, *p* = 0.01, adjusted *p* = 0.02). In particular, Hochberg’s GT2 post-hoc tests found higher anxiety scores for students attending the second year in comparison with students attending the fifth year (*p* = 0.03), and higher depression and lower mental HRQoL in supplementary years students compared with fifth year students (*p* = 0.02 and *p* = 0.02 respectively). Moreover, fifth year students also showed lower stress in comparison with both second year students (*p* = 0.02) and those attending supplementary years (*p* = 0.03).

### 3.2. Associations between Negative Emotional States, Sleep Quality, and Health-Related Quality of Life

The results of correlation analyses are shown in Table 2. All the correlations examined were significant. Not surprisingly and in line with the constructs’ characteristics, strong relationships were found between the PSQI total score and its specific components and among anxiety, depression, and stress as measured by the DASS-21. On the other hand, the correlation between physical and mental health-related quality of life was small. 

The three DASS-21 scales all showed moderate associations with overall sleep quality, as well as with the sleep disturbances and the daytime dysfunction components. Depression and stress both showed substantial associations with mental HRQoL and a somewhat weaker association with physical HRQoL, whereas the relationships between anxiety and both the PCS and the MCS were comparable and barely moderate. Similarly, moderate associations were found between the PSQI total score and both the PCS and the MCS.

Two hierarchical regression models with the PCS and the MCS as dependent variables were estimated to further explore the observed bivariate relationships, investigating in particular the association between sleep quality components (the PSQI subscores) and health-related quality of life controlling for negative emotional states. In both models, the same independent variables were subsequently inserted in three steps: sex and age in step 1, DASS-21 anxiety, depression, and stress scores in step 2, and the seven PSQI components in step 3. Coefficients and summary statistics for the final models (step 3) are shown in Table 3. Durbin-Watson statistics’ values were around 2 and variance inflation factors were below 3 for all the variables in the two variables, and thus significant multicollinearity was ruled out. Overall, the models explained 20% of the variance in the PCS scores and 40% of the variance in the MCS scores, respectively.

In the model predicting the PCS, the addition of the DASS-21 scores in step 2 produced a significant increase in R^2^ (ΔR^2^ = 0.16, *F*(3,401) = 26.13, *p* < 0.001), and so did the insertion of the sleep quality components in step 3 (ΔR^2^ = 0.04, *F*(7,394) = 2.54, *p* < 0.05). Anxiety (*p* < 0.001, β = −0.34), stress (*p* < 0.05, β = 0.14), and PSQI daytime dysfunction (*p* < 0.05, β = −0.11) showed significant associations with physical HRQoL, whereas no significant relationships were found between the remaining sleep quality components and the PCS.

In a similar way, the contributions of both the DASS-21 scales in step 2 (ΔR^2^ = 0.36, *F*(3,401) = 75.62, *p* < 0.001) and the PSQI subscores in step 3 (ΔR^2^ = 0.03, *F*(7,394) = 3.03, *p* < 0.01) were significant in the model predicting the MCS. Specifically, depression (*p* < 0.001, β = −0.39), stress (*p* < 0.001, β = −0.24), PSQI perceived sleep quality (*p* < 0.01, β = −0.14), and PSQI daily dysfunction (*p* < 0.05, β = −0.10) were significantly associated with mental HRQoL.

### 3.3. Mediation Model Examining Sleep Quality as a Mediator in the Relationship between Negative Emotional States and Health-Related Quality of Life

The examined mediation models are graphically represented in Figure 1, and total, direct, and indirect effects are summarized in Table 4. Overall, mediation analysis showed that anxiety, depression, and stress indirectly influence both physical and mental HRQoL through their effects on sleep quality (bootstrap confidence intervals for all the examined indirect effects did not encompass zero). Furthermore, higher levels of negative emotional states were associated with worse sleep quality (cf., the coefficients reported on the arrows from anxiety, depression, and stress to sleep quality in Figure 1), and participants with worse sleep quality consistently reported lower physical and mental HRQoL (cf., arrows from sleep quality to physical health-related quality of life and mental health-related quality of life in Figure 1). Depression and stress did not influence physical HRQoL independent of their effect on sleep quality (*p* = 0.27 and *p* = 0.08, respectively), whereas anxiety did not exert its effect on mental HRQoL independent of its effect on sleep quality. On the other hand, anxiety showed a significant direct effect on physical HRQoL, and depression and stress direct effects on mental HRQoL were significant too (cf., direct effect section in Table 4).

## 4. Discussion

This study sought to investigate self-reported sleep quality, negative emotional states, and health-related quality of life in a convenience sample of medical and dental students attending one of the largest Italian universities, and ultimately shed light on the purported mechanisms underlying the relationships between these outcomes, with a particular emphasis on the role of sleep.

Overall, we found that up to 62% of the students who took part in the study showed poor sleep quality and 32% reported habitually sleeping less than seven hours, i.e., less than the sleep time duration recommended for young adults by the National Sleep Foundation [21]. Together with the high observed mean sleep latency (which has been found to be a risk factor for clinical sleep disorders [57]), these findings highlights relevant sleep problems in our sample and confirm the alarming results previously reported for medical students, drawing attention at the same time to the worsening of sleep quality, which is likely attributable at least in part to the pandemic climate and the restrictive and preventive measures adopted in Italy in the period of our data collection (including mandatory use of face masks at work and in public spaces, limited access to entertainment facilities, precautionary isolation up to ten days for those who had a contact with someone infected with COVID-19, and hybrid courses with both in presence and online lessons at universities). Interestingly, this picture was found to be consistent in our sample, since no difference were observed in sleep quality across the years of study and between men and women. Hence, different study duties and responsibilities across the years (i.e., major basic sciences study load in the first three years vs. increased amount of practical experiences and clinical placements in the last three years in line with the Italian medical education curriculum [58]) were apparently not related to overall sleep quality in our sample.

Concerning the relationship between sleep quality and HRQoL, regression analyses showed significant associations among the PSQI components and both physical and mental HRQoL after controlling for demographic characteristics and negative emotional states. In particular, higher scores of daytime dysfunction were shown to be associated with lower physical and mental HRQoL, whereas worse perceived sleep quality (higher scores) was uniquely associated with lower mental HRQoL. Thus, medical students involved in this study consistently reported that the daytime dysfunction attributable to their poor sleep yielded a negative impact on their overall well-being, whereas their perception of poor sleep quality apparently exerted a specific negative effect on the psychological aspects of their functioning. These findings are indeed in line with previous research reporting significant and differential relationships between different sleep complaints and physical and mental HRQoL as measured by the SF-12 both in the general population and among university students [59,60]. However, the specific pattern of associations we found (i.e., significant relationships involving only the PSQI components of perceived sleep quality and daily dysfunction) was not observed before and might reflect the particular condition of medical students, for whom issues such as non-restorative sleep and impairment of their daytime functioning could be particularly salient.

Mediation analyses were conducted in order to further explore these associations, assuming on the basis of the findings previously mentioned that sleep might operate as a mediator in the relationship between psychological distress and HRQoL [37,38,39,40,43]. In addition to the direct significant associations between anxiety and physical HRQoL and between depression and stress and mental HRQoL, we found that sleep quality significantly mediated the relationships between anxiety, depression, and stress on one side and both physical and mental HRQoL on the other. That is, negative emotional states seemingly exerted an influence on HRQoL through their unique effects on sleep quality (with higher anxiety, depression, and stress associated with worse sleep quality, and worse sleep quality associated with worse physical and mental HRQoL). These results are essentially in line with previous studies conducted in diverse populations that considered quality of life as outcome, which found that sleep quality mediated the relationships between depression and quality of life among older Portuguese people [37], between perceived stress and physical and mental HRQoL among pregnant women [39], and between both depression and anxiety and quality of life in veterans with diabetes from the United States [38]. Conversely, the only mediation study previously conducted on medical students by Liu et al. [43] examined a somewhat different model, finding that insomnia mediated the association between perceived stress and depression. Although apparently contradictory with the model hypothesized in our study, this evidence is actually compatible with our findings. In fact, perceived stress has been defined as the degree to which an individual perceives aspects of his/her life as uncontrollable and overwhelming [61,62], and it is clearly a distinct construct from more defined negative emotional states as conceptualized and measured by the DASS-21. Rather, the DASS-21 stress scale measures a constellation of states including tension, difficulty relaxing and irritability that altogether is quite similar to generalized anxiety disorder [50,63]. Thus, given the already addressed bidirectional and recursive relationship between poor sleep and psychological distress, it is well conceivable that sleep problems mediate both the relationship between primary experienced stress and negative affect and that between the latter and overall quality of life.

In any case, the implications of our results might be noteworthy. The finding that self-reported sleep quality plays a mediating role in the relationships between negative emotional states and HRQoL even when direct associations are not observed highlights the pivotal role of sleep in health promotion and suggests that sleep assessment should be part of routine screening. Sleep quality might thus represent a main target for preventative intervention for medical students, and its improvement is likely to yield several advantages, with plausible positive effects on mental health and academic performance. Such initiatives are indeed particularly appropriate in the current period, considering the reported detrimental effect of the COVID-19 pandemic on young adults’ sleep quality and the documented association between poor sleep and COVID-19 related mental health issues [64,65,66]. To this purpose, several solutions and strategies might be implemented.

Given that medical students most likely receive inadequate sleep education [67] resulting in poor sleep awareness and knowledge [17], psychoeducational interventions might be the first step. In fact, sleep education along with sleep self-monitoring has been shown to reduce maladaptive beliefs about sleep and sleep latency among university students in a randomized controlled trial [68], but a single lecture on sleep physiology emphasizing the importance of good sleep hygiene practices failed to improve sleep quality in preclinical and clinical medical students [69]. Thus, sleep education and instructions for sleep hygiene might be enhanced by self-monitoring and other self-care strategies and low-intensity interventions drawn from behavioral medicine [70] and cognitive behavioral therapy for insomnia [71,72], which have been shown to be effective in ameliorating sleep quality and daytime functioning [73] as well as psychological distress [74,75]. The dissemination and teaching of such procedures might be easily introduced in the medical curriculum as practical experiences in line with previous proposals for improving students’ well-being [1] and could possibly produce a twofold benefit for future physicians, enhancing both their sleep quality and their clinical efficacy in dealing with sleep issues in the long-term.

Ultimately, this is the first study which explored the mediating role of sleep in the relationship between negative affect and quality of life specifically among medical students. In addition to confirming the already reported high prevalence of sleep complaints in this population, our work provides evidence regarding the possible mechanisms through which sleep might influence the overall well-being of future physicians and encourage speculations about possible strategies to address these issues. However, these promising results are preliminary and further research is needed in this field. To guide these future efforts, several limitations of this study should be thoroughly considered. In the first place, the sample included medical and dental students from a single university, with a clear majority of female participants (up to 83%). Despite that it might partially reflect the current context of medical education (according to the Italian Ministry of Education’s open data, up to 55% of the students newly enrolled in medicine undergraduate courses in 2021 were female [76], and a widespread “feminization” trend in medicine has been consistently reported in the last decades [77]), the overrepresentation of women is particularly pronounced in our sample and could possibly have affected our results given the previous evidence of higher psychological distress among female medical students [78,79]. Furthermore, the study procedure made it impossible to reliably ascertain the quality of the responses, the response rate could not be obtained, and selection bias cannot be excluded, and therefore, for all these reasons, our results might lack generalizability and might be not representative of the condition of medical students. Moreover, although mediation analysis was employed to evaluate the purported mechanisms implied in the relationships between the variables under study, the research design was cross-sectional, and thus no claims about causality can be made. In addition, several factors which could possibly have an impact on the investigated variables or could play a mediating or moderating role in their associations were not considered, including specific sleep-related conditions (e.g., excessive daytime sleepiness, obstructive sleep apnea, and other sleep disorders), diagnosed medical and/or psychiatric diseases, chronotype [80], use of prescribed and unprescribed medications, and substance use [81], as well as positive assets such as psychological resilience [82]. Moreover, the research design did not allow us to control for other plausible confounding contingencies (e.g., family stress or stress related to final exams for those responding in the latter period) that could influence the study’s main outcomes. Finally, sleep quality in this study was assessed by means of a retrospective self-report questionnaire only, despite the reliability of such instruments has been questioned in comparison with other methodologies (e.g., sleep diaries, actigraphy, or polysomnography) [83,84].

Forthcoming studies should take into account these critical aspects and evaluate the solidity of our findings on larger samples adopting robust longitudinal designs.

## 5. Conclusions

This study showed significant rates of poor sleep in a sample of medical and dental students from one of the largest Italian universities. Controlling for demographic variables and negative emotional states, sleep quality components as measured by the PSQI were found to be associated with HRQoL, with perceived sleep quality and daytime dysfunction playing a preeminent role. Furthermore, sleep quality was shown to mediate the relationships between anxiety, depression, and stress and both physical and mental HRQoL. Although further studies are needed to confirm these preliminary findings, they altogether highlight the relevance of sleep in medical students’ well-being and support the inclusion of sleep monitoring and strategies to enhance sleep quality in health promotion interventions for this population.

## Figures and Tables

**Figure 1 ijerph-20-00026-f001:**
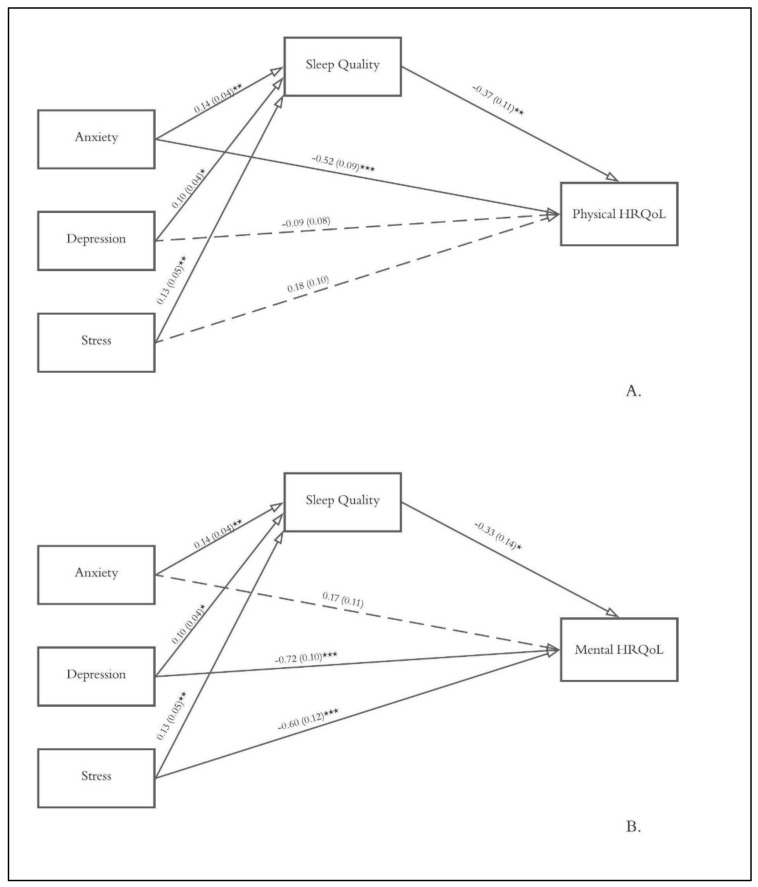
Examined mediation models with physical (**A**) and mental (**B**) health-related quality of life as consequents, anxiety, depression, and stress as antecedents, and sleep quality as mediator. Unstandardized path coefficients with standard errors are reported. Significant paths are represented with solid lines, while non-significant ones are represented with dashed lines. * *p* < 0.05; ** *p* < 0.01; *** *p* < 0.001.

**Table 1 ijerph-20-00026-t001:** Participants’ (*n* = 407) socio-demographic and sleep characteristics.

Variable	*n* (%)	Mean (SD)
Sex		
Female	339 (83.3)	
Male	66 (16.2)	
Not declared	2 (0.5)	
Age		24.2 (2.6)
Study course		
Medicine and Surgery	394 (96.8)	
Dentistry	13 (3.2)	
Study year		
First year	19 (4.7)	
Second year	38 (9.3)	
Third year	54 (13.3)	
Fourth year	48 (11.8)	
Fifth year	87 (21.4)	
Sixth year	102 (25.1)	
Supplementary years	59 (14.5)	
Total sleep time (hours)		6.9 (1.0)
Sleep onset latency (minutes)		35.7 (37.2)
Sleep efficiency index (%)		85.5 (11.2)

Note. Sleep efficiency index in percentage was obtained dividing total sleep time for hours spent in bed.

**Table 2 ijerph-20-00026-t002:** Pearson’s correlation coefficients between sleep quality components (as measured by the PSQI), anxiety, depression, stress (measured by the DASS-21), and physical (SF-12 PCS) and mental (SF-12 MCS) health-related quality of life.

Variable	Mean (SD)	1.	2.	3.	4.	5.	6.	7.	8.	9.	10.	11.	12.	13.
1. PSQI total	7.0 (3.3)	1												
2. PSQI perceived sleep quality	1.4 (0.7)	0.72 ***	1											
3. PSQI sleep latency	1.5 (1.1)	0.75 ***	0.50 ***	1										
4. PSQI sleep duration	0.8 (0.7)	0.59 ***	0.41 ***	0.27 ***	1									
5. PSQI habitual sleep efficiency	0.6 (0.8)	0.64 ***	0.32 ***	0.41 ***	0.45 ***	1								
6. PSQI sleep disturbances	1.2 (0.6)	0.60 ***	0.41 ***	0.39 ***	0.22 ***	0.19 ***	1							
7. PSQI sleep medication use	0.3 (0.8)	0.53 ***	0.23 ***	0.26 ***	0.15 **	0.16 **	0.22 ***	1						
8. PSQI daytime disfunction	1.1 (0.7)	0.51 ***	0.33 ***	0.20 ***	0.17 **	0.14 **	0.35 ***	0.19 ***	1					
9. DASS-21 anxiety	6.8 (4.8)	0.41 ***	0.24 ***	0.24 ***	0.17 **	0.16 **	0.48 ***	0.22 ***	0.39 ***	1				
10. DASS-21 depression	9.1 (5.4)	0.40 ***	0.25 ***	0.27 ***	0.19 ***	0.20 ***	0.36 ***	0.17 **	0.39 ***	0.60 ***	1			
11. DASS-21 stress	12.8 (4.7)	0.42 ***	0.35 ***	0.25 ***	0.19 ***	0.14 **	0.43 ***	0.19 ***	0.37 ***	0.63 ***	0.66 ***	1		
12. SF-12 PCS	53.8 (7.2)	−0.29 ***	−0.21 ***	−0.17 ***	−0.10 *	−0.11 *	−0.29 ***	−0.17 **	−0.28 ***	−0.39 ***	−0.27 ***	−0.22 ***	1	
13. SF-12 MCS	33.6 (10.0)	−0.35 ***	−0.31 ***	−0.19 ***	−0.15 **	−0.20 ***	−0.28 ***	−0.11 *	−0.35 ***	−0.37 ***	−0.57 ***	−0.54 ***	−0.10 *	1

Note. PSQI: Pittsburgh Sleep Quality Index; DASS-21: 21-item Depression Anxiety Stress Scale; SF-12 PCS: Short Form-12 Physical Component Summary; SF-12 MCS: Short Form-12 Mental Component Summary; * *p* < 0.05; ** *p* < 0.01; *** *p* < 0.001.

**Table 3 ijerph-20-00026-t003:** Hierarchical multiple regression models (final models) with physical (SF-12 PCS) and mental (SF-12 MCS) health-related quality of life as dependent variables.

	SF-12 PCS			SF-12 MCS		
	Coefficient (SE)	t	Beta	Coefficient (SE)	t	Beta
Sex	−1.17 (0.85)	−1.38	−0.06	1.26 (1.02)	1.22	0.05
Age	−0.18 (0.13)	−1.42	−0.07	−0.04 (0.15)	−0.25	−0.01
DASS-21 anxiety	−0.50 (0.10)	−5.20 ***	−0.34	0.16 (0.12)	1.39	0.08
DASS-21 depression	−0.08 (0.09)	−0.94	−0.06	−0.72 (0.10)	−6.96 ***	−0.39
DASS-21 stress	0.22 (0.11)	2.07 *	0.14	−0.53 (0.13)	−4.14 ***	−0.24
PSQI perceived sleep quality	−0.83 (0.62)	−1.33	−0.08	−2.09 (0.75)	−2.79 **	−0.14
PSQI sleep latency	−0.01 (0.38)	−0.03	−0.00	0.64 (0.46)	1.38	0.07
PSQI sleep duration	0.38 (0.59)	0.64	0.03	0.77 (0.71)	1.07	0.05
PSQI habitual sleep efficiency	−0.14 (0.47)	−0.29	−0.02	−1.11 (0.57)	−1.94	−0.09
PSQI sleep disturbances	−10.6 (0.73)	−1.47	−0.08	−0.08 (0.89)	−0.09	−0.01
PSQI sleep medication use	−0.49 (0.40)	−1.21	−0.06	0.37 (0.49)	0.75	0.03
PSQI daytime dysfunction	−1.24 (0.57)	−2.17 *	−0.11	−1.53 (0.69)	−2.2 *	−0.10
Summary statistics						
Model F	8.41 ***			22.14 ***		
R^2^	0.20			0.40		
Adjusted R^2^	0.18			0.39		

Note. PSQI: Pittsburgh Sleep Quality Index; DASS-21: 21-item Depression Anxiety Stress Scale; SF-12 PCS: Short Form-12 Physical Component Summary; SF-12 MCS: Short Form-12 Mental Component Summary; Beta: standardized regression coefficient. * *p* < 0.05; ** *p* < 0.01; *** *p* < 0.001.

**Table 4 ijerph-20-00026-t004:** Total, direct, and indirect effects (unstandardized estimates) of the mediation models examining the association between negative affect (anxiety, depression, and stress) and physical and mental health-related quality of life with sleep quality as mediator.

	Estimate (SE)	t ^a^	95% CI ^b^
**Total effect**			
DASS-21 anxiety → PCS	−0.57 (0.09)	−6.17 ***	[−0.75, −0.39]
DASS-21 depression → PCS	−0.13 (0.08)	−1.52	[−0.30, 0.04]
DASS-21 stress → PCS	0.13 (0.10)	1.27	[−0.07, 0.33]
DASS-21 anxiety → MCS	0.13 (0.11)	1.12	[−0.09, 0.35]
DASS-21 depression → MCS	−0.70 (0.10)	−7.34 ***	[−0.96, −0.55]
DASS-21 stress → MCS	−0.64 (0.12)	−5.18 ***	[−0.88, −0.40]
**Direct effect**			
DASS-21 anxiety → PCS	−0.52 (0.09)	−5.63 ***	[−0.70, −0.34]
DASS-21 depression → PCS	−0.09 (0.08)	−1.10	[−0.26, 0.07]
DASS-21 stress → PCS	0.18 (0.10)	1.75	[−0.02, 0.38]
DASS-21 anxiety → MCS	0.17 (0.11)	1.50	[−0.05, 0.39]
DASS-21 depression → MCS	−0.72 (0.10)	−7.01 ***	[−0.93, −0.52]
DASS-21 stress → MCS	−0.60 (0.12)	−4.80 ***	[−0.84, −0.35]
**Indirect effect**			
DASS-21 anxiety → PSQI → PCS	−0.05 (0.02)		[−0.10, −0.01]
DASS-21 depression → PSQI → PCS	−0.04 (0.02)		[−0.08, −0.003]
DASS-21 stress → PSQI → PCS	−0.05 (0.02)		[−0.10, −0.01]
DASS-21 anxiety → PSQI → MCS	−0.04 (0.02)		[−0.10, −0.01]
DASS-21 depression → PSQI → MCS	−0.03 (0.02)		[−0.07, −0.001]
DASS-21 stress → PSQI → MCS	−0.04 (0.02)		[−0.10, −0.004]

Note. PSQI: Pittsburgh Sleep Quality Index; DASS-21: 21-item Depression Anxiety Stress Scale; PCS: Short Form-12 Physical Component Summary; MCS: Short Form-12 Mental Component Summary. *** *p* < 0.001. ^a^: t statistics and relative p-values were computed only for total effects and direct effects. ^b^: 95% percentile bootstrap confidence intervals are reported for indirect effects.

## Data Availability

The data presented in this study are available from the corresponding author upon reasonable request.

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
