# Peer review of "The Mediating Role of Sleep Quality in the Relationship between Negative Emotional States and Health-Related Quality of Life among Italian Medical Students"

_ijerph, 2022, doi:10.3390/ijerph20010026_

Round 1

Reviewer 1 Report

Interesting study. Impressive work on statistical analysis.  

There are some major shortcomings to this study that have been, for the most part, discussed in the Discussion section including the sole use of subjective questionnaires without any objective measures.

It is not clear if the participants had any diagnosed pathology or were taking any prescribed or OTC medications that could have affected their sleep or other measures used in this study.

Please explain why both Cronbach’s alpha and McDonald’s omega were used for evaluating measures’ reliability.

Did the authors consider any potential confounding factors? For example, if the participants were going through a stressful period such as final exams could this not have affected both their mood, stress level and sleep quality? Please discuss.

Text contains minor grammatical errors that need to be corrected.

Author Response

We thank the reviewer for his/her appreciation and thorough observations that we find helpful in ameliorating our work. Below we report our responses to the reviewer's comments (reviewer's comments are reported in bold). Changes were made in the revised manuscript file using the Track Changes function of Microsoft Word and were highlighted in yellow.

  • It is not clear if the participants had any diagnosed pathology or were taking any prescribed or OTC medications that could have affected their sleep or other measures used in this study.

Indeed, we did not control for medical or psychiatric conditions or comorbidities and medical drug consumption (besides sleep medication use as assessed with the PSQI) in our study. We explicitly addressed this point in the revised manuscript in the paragraph dealing with the limitations of our study in which we already mentioned the possible consequences of not investigating diagnosed sleep disorders.

  • Please explain why both Cronbach’s alpha and McDonald’s omega were used for evaluating measures’ reliability.

There is growing methodological evidence that McDonald's omega coefficient (which is in fact a more general form of Cronbach's alpha with less strict assumptions about items data distribution) might represent a more consistent estimate of reliability in comparison with Cronbach's coefficient (e.g., Hayes & Coutts, 2020). However, since Cronbach's alpha is more commonly reported (and the validation studies of the instruments we used indeed reported alpha coefficients as reliability estimates) we chose to report both omega and alpha (for which we observed only minimal discrepancies in our sample) in order to allow a comparison with previous results.

  • Did the authors consider any potential confounding factors? For example, if the participants were going through a stressful period such as final exams could this not have affected both their mood, stress level and sleep quality? Please discuss.

This is an interesting point which we definitely took into account in designing our study. As reported, the participants answered the survey between March and July. This timeframe was chosen in order to avoid the interference of pre-exam distress (exams for medical students are usually held from late june/july to september in our university). Actually, it is well possible that those who answered closer to the final exams were experiencing a more stressful period, but we were not able to control for this circumstance (and furtherly reducing the timeframe to access the survey would have probably sigificantly decreased our sample size). We acknowledged this possible confounding factor in the revised manuscript in the section dealing with the study's limitations.

  • Text contains minor grammatical errors that need to be corrected.

We thank the reviewer for noticing. We conducted additional grammar check on the manuscript and corrected several misprints.

Reviewer 2 Report

This interesting manuscript relates to medical students’ mental health, quality of sleep, and quality of life.

The manuscript has a suitable introduction.

The method is appropriately described. Although it is recommended to include a brief description of the courses in the mentioned years. It is necessary to recognize which are the preclinical years and which are the clinical years (with practices in hospitals). In addition, it is necessary to briefly describe the existing restrictions due to the COVID-19 epidemic at the time of data collection, for example, in strict lockdown; as well as the measures implemented by the university, for example, hybrid courses (face-to-face-virtual), only virtual, etc. On the other hand, it is necessary to report the number of students to whom the questionnaire was sent and what was the response rate. It is necessary to mention the percentage of men and women enrolled in the medicine and dentistry courses of the referred university during the evaluation period.

In the limitations, reflect more on the possible bias in the results due to the overrepresentation of women. It is known that women are more susceptible to depression, anxiety, and stress compared to men.

Author Response

We thank the reviewer for his/her useful observations and advice which gave us some helpful hints in revising the manuscript. Below we report our responses to the reviewer's comments (excerpts from the reviewer's report are reported in bold). Changes were made in the revised manuscript file using the Track Changes function of Microsoft Word and were highlighted in yellow.

  • The method is appropriately described. Although it is recommended to include a brief description of the courses in the mentioned years. It is necessary to recognize which are the preclinical years and which are the clinical years (with practices in hospitals).

We acknowledge that this point was not clarified in the submitted version of the manuscript. In fact, in the Italian education systems, degree courses in medicine and dentistry are not officially split in preclinical and clinical years, and clinical experiences are distributed throughout the six years of study. However, there is a major concentration of practical and clinical activities in the last three years, with slightly increasing hospital duties. We mentioned this characteristic of the study course in the revised manuscript in the Discussion section and added in the references an article authored by Consorti, Familiari, Lotti & Torre (2021) that describes the Italian medical curriculum in-depth.

  • In addition, it is necessary to briefly describe the existing restrictions due to the COVID-19 epidemic at the time of data collection, for example, in strict lockdown; as well as the measures implemented by the university, for example, hybrid courses (face-to-face-virtual), only virtual, etc.

We definitely agree that this information is relevant since we mentioned the possible impact of the restrictive measures adopted in Italy during our data collection. Indeed, in that period it was mandatory to wear face masks at work and in public spaces, access to entertainment facilities was limited, and up to ten days of precautionary isolation were prescribed to those who had a close contact with subjects infected with Covid-19, whilst university courses were held both in presence and online for those unable to participate (hybrid). We made this explicit in the Dicussion section in the revised manuscript.

  • On the other hand, it is necessary to report the number of students to whom the questionnaire was sent and what was the response rate. It is necessary to mention the percentage of men and women enrolled in the medicine and dentistry courses of the referred university during the evaluation period.

As we mentioned in the paragraph dealing with our study's shortcomings, it was not possible to obtain the exact number of students to whom the questionnaire was delivered and thus to estimate the response rate for the survey. We acknowledge that this is significant limitation and sought to highlight it in the Discussion section. Concerning the percentage of men and women enrolled in medical and dental courses in our university, this data is actually not accessible in public data repository. However, in the revised manuscript we further examined the women-men imbalance in our sample reporting the percentage of women enrolled in medicine undergraduate courses in Italy in 2020 according to Italian Ministry of Education open data.

  • In the limitations, reflect more on the possible bias in the results due to the overrepresentation of women. It is known that women are more susceptible to depression, anxiety, and stress compared to men.

Interesting and relevant point. As mentioned above, we further discussed the overrepresentation of women in the revised manuscript in light of public available data and the reported feminazation trend in medicine. We also added a couple of reference (Burger & Scholtz, 2018, Dahlin et al., 2005) concerning the purported high levels of psychological distress among female medical students.

Reviewer 3 Report

Overall the manuscript clearly describes a study into the mediating role of sleep quality on the association between negative emotional states and quality of life in medical studies. Although the study populations may be fairly specific and reduce generalizability of the results, the study is relevant. There are some issues with the study that need to be addressed.

1)      It is not clear why dentistry students were included in this study. Although the reviewer does not have a lot of knowledge about the difference in the study program of both majors in the Italian system, it seems that the program and responsibilities for both may be different over time and thus especially since year of study is included in the analysis this seems problematic. And since the amount of dentistry students is really low (13), including this as a covariate in the analysis does not seem feasible. I would suggest excluding this group from the analyses.

2)      It would be informative to include some information on the medical education in Italy. Since the education is different in different countries, it would be helpful to have some knowledge on the medical education (responsibilities etc) in the different years of study. It is for example not clear what supplemental education entails (to a non Italian).

3)      Line 243, I would specify here that daytime function refers to the PSQI daytime function subscale, as it gets confusing with the SF-12 as outcome variable (fairly close to the same concept).

4)      Figure 1: The arrows in the figure suggest a know direction of the associations, but since this was measured in the cross-sectional study, directionality and causality cannot be determined.

5)      Line 289: The authors imply here that the seemingly higher percentage of students scoring above cut-off on the PSQI compare to previous studies is related to the covid pandemic. However, since previous studies were performed in different populations, there could be various reasons for potential differences with other studies.

Author Response

We thank the reviewer for his/her thorough observations and comments. We sought to take advantage from his/her keen suggestions in revising and ameliorating the manuscript. Below we report our point-by-point responses to the reviewer's comments (the reviewer's points are reported in bold). Changes were made in the revised manuscript file using the Track Changes function of Microsoft Word and were highlighted in yellow.

  • It is not clear why dentistry students were included in this study. Although the reviewer does not have a lot of knowledge about the difference in the study program of both majors in the Italian system, it seems that the program and responsibilities for both may be different over time and thus especially since year of study is included in the analysis this seems problematic. And since the amount of dentistry students is really low (13), including this as a covariate in the analysis does not seem feasible. I would suggest excluding this group from the analyses.

We repeateadly reflected on this problematic aspect of our study and we thank the reviewer for pointing it out. Originally, we chose to survey both medical and dentistry students given that their study curricula and clinical duties in hospital over the six years of course are very similar in Italian universities and also considering that these two students' populations have been surveyed together in previous studies (e.g., Aboalshamat et al., 2015, Ey, Henning & Shaw, 2000, Henning, Ey & Shaw, 1998). We also sought to examine possible differences among the two groups considering the study course (medicine vs dentistry) as a covariate in the analyses. However, as the reviewer noticed, this what not possible due to the very low sample size for dentistry students (although our numbers indeed reflect the low proportion of dentistry students over medical students in our university, which enrolls at maximum approximately 60 dentistry students and 1000 medical students every year according to information retrieved on its website). Despite we failed in reaching the adequate sample size for a comparison, we chose to include dentistry students in the final sample anyway, in consideration of the above reported arguments.

  • It would be informative to include some information on the medical education in Italy. Since the education is different in different countries, it would be helpful to have some knowledge on the medical education (responsibilities etc) in the different years of study. It is for example not clear what supplemental education entails (to a non Italian).

We understand that there are many differences in medical education across countries, sometimes even concerning the core study curriculum. We appreciate this comment and a similar suggestion made by another reviewer. In Italy, medicine and dentistry degree courses lasts six years. Practical and clinical activities are distributed over the whole study period and a neat separation between preclinical and clinical years is not establihed, but clinical placements and hospital duties increase over the last three years. Although a detailed description of medical education in Italy is likely out of the scope of our paper, in the revised manuscript we remarked this last point (more clinical activities in the last years of study) in the Discussion sections and added in the references a recent article authored by Consorti, Familiari, Lotti & Torre (2021) that describes the Italian medical curriculum in-depth.

  • 3) Line 243, I would specify here that daytime function refers to the PSQI daytime function subscale, as it gets confusing with the SF-12 as outcome variable (fairly close to the same concept).

We thank the reviewer for noticing. We specified that both perceived sleep quality and daytime dysfunction are indeed PSQI subscores in the revised manuscript for the model predicting the SF-12 PCS and the model predicting the SF-12 MCS by indicating 'PSQI perceived sleep quality' and 'PSQI daytime dysfunction'.

  • 4) Figure 1: The arrows in the figure suggest a know direction of the associations, but since this was measured in the cross-sectional study, directionality and causality cannot be determined.

Interesting and compelling point. In the paragraph dealing with our study's limitations in the Discussion section, we acknowledged that the cross-sectional design we adopted does not allow to draw conclusions concerning the directionality of the examined associations and we do definitely agree with the reviewer concerning this methodological point. Despite the debate is still intense, we also retain that longitudinal designs are necessary to corroborate causality claims involving mechanisms that may express over time. That said, in our mediation analyses we relied on the approach outlined by Hayes (2018, reference in the text) and MacKinnon (2008). Conventionally, mediation diagrams are represented with directional arrows from the antecedent(s) variable(s) pointing to the mediator(s) and from the mediator(s) to the consequent(s) variable(s) and we adopted the same graphical format adopted by Hayes and MacKinnon (for both cross-sectional and longitudinal mediation models). This also reflects the fact that a theoretical hypothesis concerning the direction of the explored relationships is implicit in examining a mediation model (and indeed, the indirect effect of X on Y through M is naturally diverse from the indirect effect of Y on X through M). Competitive models (such as a model exploring sleep quality as antecedent and negative emotional states as parallel mediatiors) might be tested to examine alternative hypotheses. Thus, we presented the arrows in the mediation diagram to convey our hypothesis about the mechanisms involved in the relationship between the investigated variables and to make clear that the indirect effects we computed are obtained by multiplying the regression coefficients of negative emotional states on sleep quality for the regression coefficients of sleep quality on physical and mental HRQoL. Anyway, by pointing out the cross-sectional nature of our research we also tried to make explicit that we do not interpret our results as a robust proof of actual directional relationships.

  • 5) Line 289: The authors imply here that the seemingly higher percentage of students scoring above cut-off on the PSQI compare to previous studies is related to the covid pandemic. However, since previous studies were performed in different populations, there could be various reasons for potential differences with other studies.

We do agree with the reviewer that the pandemic climate is just one of the possible explanations for the high percentage of students reporting poor sleep quality in our study. We chose to remark its possible role explicitly to highlight a major trend in the literature (namely, the impact of the Covid-19 pandemic on sleep health and psychological well-being), but we acknowledge that the sentence in the submitted manuscript was too strong and could be misinterpreted. We changed it in "[...] drawing attention at the same time to the worsening of sleep quality, which is likely attributable at least in part to the pandemic climate [...]" (part added in bold).